# Learning Temporally Invariant and Localizable Features via Data Augmentation for Video Recognition

**Abstract.** Deep-Learning based video recognition has shown promising improvements along with the development of large-scale datasets and spatio-temporal network architectures. In image recognition, learning spatially invariant features is a key factor for improving recognition performance and robustness. Data augmentation based on visual inductive priors such as crop, flip, rotation, or photometric jittering is a representative approach to achieve these features. Recent state-of-the-art recognition solutions are relied on modern data augmentation strategies that exploit mixture of augmentation operations. In this study, we extend these strategies to the temporal dimension for videos to learn temporally invariant, or temporally localizable features to cover temporal perturbations, or complex actions in videos. Based on our novel temporal data augmentation algorithms, video recognition performances are improved in a limited amount of training data, compared to spatial-only data augmentation algorithms, including the 1st Visual Inductive Priors (VIPriors) for data-efficient action recognition challenge. Furthermore, learned features are temporally localizable that cannot be achieved from the spatial augmentation algorithms.

## 1 Introduction

A lot of augmentation techniques have been proposed to increase recognition performance and robustness for an environment of limited training data, or to prevent overconfidence and overfitting of large-scale data such as ImageNet [23]. These techniques can be categorized into data-level augmentation [24, 33, 8, 29, 9, 18, 10, 34], data-level mixing [52, 50, 49, 27, 42, 26], and in-network augmentation [37, 13, 19, 12, 48, 21, 41]. Data augmentation is an important component for recent state-of-the-art self-supervised learning [16, 4, 31], semi-supervised learning [45, 2, 1, 35], self-learning [46], and generative models [51, 53, 54, 20] because of its ability to learn invariant features.

Purpose of data augmentation in image recognition is to enhance generalization ability via learning spatially invariant features. Augmentations such as geometric (crop, flip, rotation, *etc.*) and photometric (brightness, contrast, color, *etc.*) transformations can model uncertain variances in a dataset. Recent algorithms have shown state-of-the-art performances in terms of complexity-accuracy

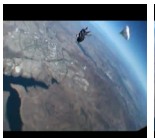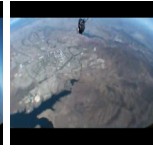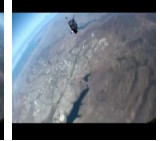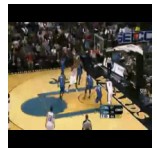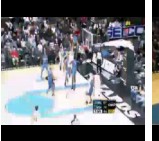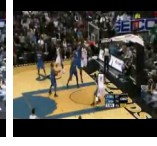

Fig. 1: Example clips of temporal perturbations. *Left*: Geometric perturbation across frames in sky-diving video due to extreme camera and object movements. *Right*: Photometric perturbation across frames in the basketball stadium due to camera flashes.

tradeoff [29, 9], or robustness [17, 18]. Some approaches [50, 49] learn localizable features that can be used as transferable features to the localization-related tasks such as object detection and image captioning. They learn simultaneously what to and where to focus on for the recognition.

Despite evolving through numerous algorithms in image recognition, there are little explorations of data augmentation and regularization in video recognition. In videos, temporal variations and perturbations should be considered as well as spatial ones. For example, Fig. 1 depicts temporal perturbations across frames in a video. This perturbation can be one of the geometric perturbations such as translation, rotation, scale, *etc.*, or the photometric perturbations such as brightness, contrast, *etc..* To handle these perturbations, not only well-studied spatial augmentations but also temporally varying data augmentations should be considered generally.

In this paper, we propose several extensions toward temporal robustness. More specifically, temporally invariant and localizable features can be modeled via data augmentations. We extend such examples of well-studied recent spatial augmentation techniques: data-level augmentation and data-level mixing. To the best of our knowledge, it is very first study that deeply analyzes temporal perturbation modeling via data augmentation in video recognition.

Contribution of this paper is summarized as follows:

– We propose an extension of RandAugment [9], called RandAugment-T, as data-level augmentation for video recognition. It can model temporally varying level of augmentation operations.
– We propose temporal extensions of CutOut [10], MixUp [52], and Cut-Mix [50] as examples of deleting, blending, and mixing data samples. Considering temporal dimension improves recognition performances and temporal localization abilities.
– Recognition results of proposed extensions on UCF-101 [36] subset for 1st Visual Inductive Priors (VIPriors) for data-efficient action recognition challenge, and HMDB-51 [25] dataset show performance improvements compared to spatial-only versions in a simple baseline.

## 2    Related Works

### 2.1    Data augmentation

**Data-level augmentation** In the beginning, to enlarge the generalization performance of a dataset and to reduce overfitting problem of preliminary networks, various data augmentation methods such as rotate, flip, crop, color jitter [23], and scale jittering [33] are proposed. CutOut [10] deletes a square-shaped box at random location to encourage the networks focus on various properties of image, not rely on the most discriminative regions. Hide-and-Seek [34] is a similar approach, but it deletes multiple regions that are sampled from the grid patches.

Recently, the methodology of combining more than one augmentation operations has been proposed. Cubuk *et al.* [8] propose a reinforcement learning-based approach to search the optimal data augmentation policy in a given dataset. However, since the search space is too large, it requires extensive time to find the optimal policy. Although an approach to mitigate this problem is proposed [29], it is still hard and time-consuming to get to the optimal augmentation strategy. To solve this, Cubuk *et al.* [9] propose RandAugment that randomly sample augmentation operations from the candidate list and cascade them. Hendrycks *et al.* [18] propose an approach called AugMix that blend images parallelly that are augmented by the operations sampled from set of candidates like RandAugment.

These techniques can model uncertain spatial perturbations such as geometric transform, photometric transform, and both of them. Since researches have focused on static images, applying these approaches into videos is a straightforward extension.

**Data-level mixing** Together with data augmentation algorithms, augmentation strategies using multiple samples have been proposed. Zhang *et al.* [52] propose an approach to manipulate images with more than one image, called MixUp. They make a new sample by blending two arbitrary images and interpolate their one hot ground-truth labels. This encourages the model to behave linearly in-between training examples. CutMix [50] combines the concepts of CutOut and MixUp, taking the best of both worlds. It replaces square-shaped deleted region in CutOut with a patch from another image. This encourages the model to learn not only what to recognize, but also where to recognize. It can be interpreted as spatially localizable feature learning. Inspired by CutMix, several methods to increase the generality have been proposed. CutBlur [49] proposed CutMix-like approach to solve the restoration problem using mixing between low-resolution and high-resolution images. They also proposed CutMixUp that is a combination of MixUp and CutMix. CutMixUp blends the two images in the one of the mask of CutMix to relax extreme changes in boundary pixels. Attribute Mix [27] uses the masks of any shape, not only squre-shaped mask. Attentive CutMix [42] also discards the square-shaped mask. It uses multiple patches sampled from the grid, and replaces the regions with another image. Smoothmix [26] focus on the 'strong edge' problem caused by the boundary of masks.

Although numerous data manipulation methods including deleting, blending, and mixing, successfully augment many image datasets, their ability when applied to video recognition to learn temporally invariant and localizable features, is not explored yet.

**In-network augmentation** Apart from the data-level approaches, several researches have proposed in-network augmentation algorithms. They usually design stochastic networks to augment in the feature-level in order to reduce predictive variance and to learn more high-level augmented features rather than to learn features from the low-level augmentations. Dropout [37] is a very first approach to regularize the overfitted models. Other approaches such as Drop-Block [13], Stochastic depth [19], Shake-Shake [12] and ShakeDrop [48] regularizations have been proposed. Manifold-MixUp [41] propose mixing strategy like MixUp, but in the feature space. The most similar approach with this study is a regularization method for video recognition, called Random Mean Scaling [21]. It randomly adjusts spatio-temporal feature in the video networks. In contrast, our approaches focus on data-level manipulations and extending from the spatial-only algorithms into temporal worlds.

## 2.2    Video recognition

For video action recognition, like image recognition area, various architectures have been proposed to capture spatio-temporal features from videos. In [39], Tran *et al.* proposed C3D that extracts features containing objects, scenes and action information through 3D convolutional layers, and then simply passed through a linear classifier. In [40], a (2+1)D convolution that focuses on layer factorization rather than 3D convolution is proposed, which is composed with 2D spatial convolution followed by 1D temporal convolution. In addition, non-local block [44] and GloRe module [6] are suggested to capture long range dependencies via self-attention and graph-based modules. By plugging them into 3D ConvNet, the network can learn long distance relations in both space and time. Another approach is two stream architectures [43, 38, 32]. In [3], a two-stream 3D ConvNet inflated from deep image classification network and pre-trained features is proposed and achieves state-of-the-art performance by pre-training it with Kinetics dataset, a large-scale action recognition dataset. Based on this architecture, Xie *et al.* [47] combined a top-heavy model design, temporally separable convolution, and spatio-temporal feature gating blocks to make low-cost and meaningful features. Recently, SlowFast [11] networks that consists of a slow path for semantic information and a fast path for rapidly changing motion information, show competitive performance with different frame rate sampling strategy. In addition to this, RESOUND [28] proposed a method to reduce the static bias of the dataset, an Octave convolution [5] is proposed to reduce spatial redundancy by dividing the frequency of features, and debiasing loss function [7] is proposed to mitigate the strong scene bias of the networks and focus on the actual action information.

Since the advent of the large-scale Kinetics dataset, most action recognition studies have pre-trained the backbone on Kinetics, which guarantees basic performances. However, based on the results of [15], architectures with large numbers of parameters are significantly overfitted when learning from the scratch on relatively small dataset such as UCF-101 [36] and HMDB-51 [25]. It indicates that training without a pre-trained backbone is a challenging issue. Compared to existing researches that have been focused on novel dataset and architectures, we focus on the regularization techniques such as data augmentation, to prevent overfitting via learning invariance and robustness in terms of spatially and temporally.

## 3  Methods

### 3.1  Data-level temporal data augmentations

First, we extend existing RandAugment [9] for video recognition. RandAugment has two hyper-parameters to optimize. One is the number of augmentation operation to apply, N, and the other is the magnitude of the operation, M. Grid search of these two parameters in a given dataset produces state-of-the-art performances in image recognition.

```python
def randaugment_T(X, N, M1, M2):
    """Generate a set of distortions.

    Args:
    X: Input video (T x H x W)
    N: Number of augmentation transformations
    to apply sequentially.
    M1, M2: Magnitudes for both temporal ends.
    """

    ops = np.random.choice(transforms, N)
    M = np.linspace(M1, M2, T)
    return [[op(X, M[t]) for t in range(T)]
        for op in ops]
```

Fig. 2: Python code for RandAugment-T based on numpy.

For video recognition, RandAugment is directly applicable to every frame of an video, however, this limits temporal perturbation modeling. To cover temporally varying transformations, we propose RandAugment-T that linearly interpolates between two magnitudes from the first frame to the last frame in a video clip. Pseudo-code of RandAugment-T is described in Fig. 2. It receives three hyper-paremeters: N, M1, and M2. N is the number of operations, which is same as RangAugment. M1 and M2 indicate magnitudes for both temporal ends, which can be any combination of levels. Set of augmentation operations (`transforms` in Fig. 2) is identical with RandAugment. However, `rotate`, `shear-x`, `shear-y`, `translate-x`, and `translate-y` can model temporally varying geometric transformations such as camera movement or object movement (Fig. 3(a)), and `solarize`, `color`, `posterize`, `contrast`, `brightness`, and `sharpness` can model photometric transformations such as brightness or contrast change due to auto-shot mode in a camera (Fig. 3(b)). Remained operations (`identity`, `autoContrast`, and `equalize`) have no magnitudes that are applied to evenly across frames.

### 3.2  Data-level temporal deleting, blending, and mixing

Regularization techniques for image recognition such as CutOut [10], MixUp [52], and CutMix [50] can be applied identically across frames in a video. CutMixUp

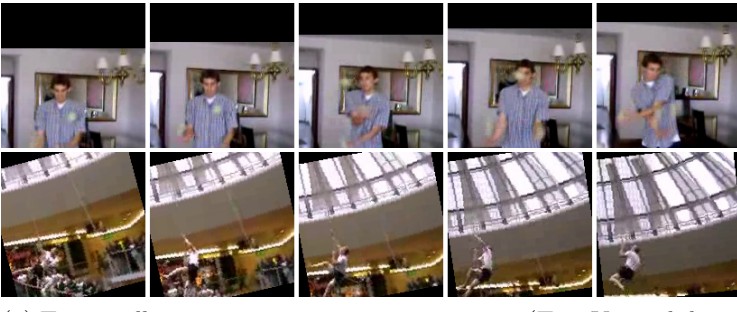

(a) Temporally varying geometric augmentations (Top: Vertical-down translation, Bottom: Clockwise rotation)

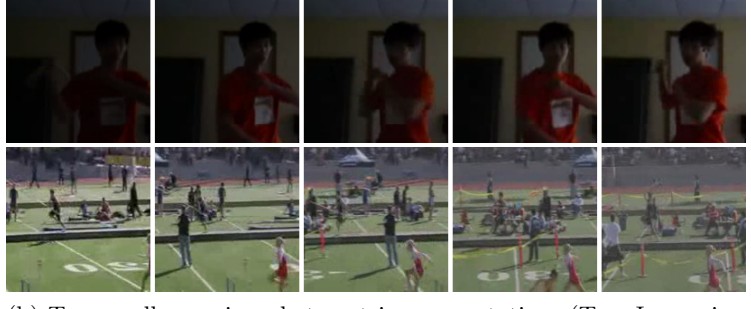

(b) Temporally varying photometric augmentations (Top: Increasing brightness, Bottom: Decreasing contrast)

Fig. 3: Example of temporally varying data augmentation operations for RangAugment-T

is a combination of MixUp and CutMix, which is proposed in [49] also can be applied to recognition to relax the unnatural boundary changes.

In this section, we propose temporal extensions of above algorithms. Frame-CutOut and CubeCutOut is the temporal and spatio-temporal extension of CutOut (Fig 4 (a)), respectively. CutOut encourages the network to better utilize the full context of the image, rather than relying on a small portion of specific spatial regions. Similarly, FrameCutOut encourages the network to better utilize the full temporal context, and the full spatio-temporal context by CubeCutOut.

FrameCutMix and CubeCutMix is the extension of CutMix [50] (Fig 4 (b)). CutMix is designed for learning of spatially localizable features. Cut and paste mixing between two images encourages the network to learn where to recognize. Similarly, FrameCutMix and CubeCutMix is designed for learning of temporally and spatio-temporally localizable features in a video. Like CutMix, mixing ratio $\lambda$ is sampled from beta distribution $Beta(\alpha, \alpha)$, where $\alpha$ is a hyper-parameter, and locations for random frames or random spatio-temporal cubes are selected based on $\lambda$.

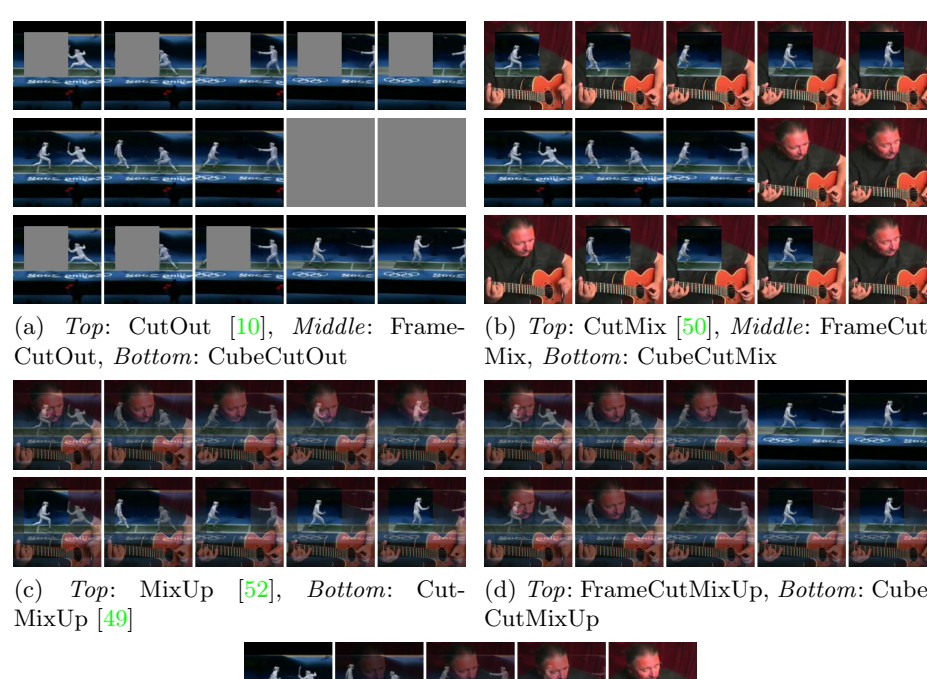

(a) *Top*: CutOut [10], *Middle*: Frame-CutOut, *Bottom*: CubeCutOut

(b) *Top*: CutMix [50], *Middle*: FrameCut-Mix, *Bottom*: CubeCutMix

(c) *Top*: MixUp [52], *Bottom*: Cut-MixUp [49]

(d) *Top*: FrameCutMixUp, *Bottom*: Cube-CutMixUp

(e) FadeMixUp

Fig. 4: Visual comparison of data-level deleting, blending, and mixing for videos. Desired ground-truth labels are calculated by the ratio of each class: *Fencing* and *PlayingGuitar*.

Like CutMixUp [49], which is the unified version of MixUp [52] and Cut-Mix [50], FrameCutMixUp and CubeCutMixUp can be designed in similar way (Fig 4 (c) and (d)) to relax extreme boundary changes between two samples. For these blend+mix algorithms, MixUp is applied between two data samples by mixing ratio $\lambda_1$, and the other hyper-parameter $\lambda_2$ is sampled from $Beta(2, 2)$. Based on $\lambda_2$, region mask $\mathbf{M}$ is selected randomly like CutMix to mix again between MixUp-ed sample and one of the two samples. Mixed data and desired ground-truth labels are formulated as below.

$$\tilde{x} = \begin{cases} (\lambda_1 x_A + (1 - \lambda_1)x_B) \odot \mathbf{M} + x_A \odot (\mathbf{1} - \mathbf{M}) & \text{if } \lambda_1 < 0.5 \\ (\lambda_1 x_A + (1 - \lambda_1)x_B) \odot \mathbf{M} + x_B \odot (\mathbf{1} - \mathbf{M}) & \text{if } \lambda_1 \geq 0.5 \end{cases}$$

$$\tilde{y} = \begin{cases} (\lambda_1 \lambda_2 + (1 - \lambda_1))y_A + (1 - \lambda_1)\lambda_2 y_B & \text{if } \lambda_1 < 0.5 \\ \lambda_1 \lambda_2 y_A + (1 - \lambda_1 \lambda_2)y_B & \text{if } \lambda_1 \geq 0.5 \end{cases}$$

(1)

where $\tilde{x}$, $\tilde{y}$, and $\odot$ indicate mixed data, modified label, and element-wise multiplication, respectively.

Table 1: Comparison between deleting, blending, and mixing frameworks.

| Type | Delete | | | Mix | | | Blend | | Blend + Mix | | |
|---|---|---|---|---|---|---|---|---|---|---|---|
| Name | CutOut [10] | Frame CutOut | Cube CutOut | CutMix [50] | Frame CutMix | Cube CutMix | MixUp [52] | Fade MixUp | CutMixUp [49] | Frame CutMixUp | Cube CutMixUp |
| Axis Spatial | ✓ | | ✓ | ✓ | | ✓ | | | ✓ | | ✓ |
| Temporal | | ✓ | ✓ | | ✓ | ✓ | | ✓ | | ✓ | ✓ |

Finally, we propose another extension of MixUp, called FadeMixUp. Inspired by fade-in, fade-out, dissolve overlap effects in videos, from MixUp, mixing ratio is smoothly changing along with temporal frames (Fig 4 (e)). In FadeMixUp, list of the mixing ratio $\tilde{\lambda}_t$ of a frame $t$, is calculated by linear interpolation between $\lambda - \gamma$ and $\lambda + \gamma$, where $\lambda$ is the mixing ratio of MixUp and $\gamma$ is sampled from $Uniform(0, min(\lambda, 1-\lambda))$. Because the adjustments of mixing ratio at the both ends are symmetric, label is same as MixUp.

$$\tilde{x}_t = \tilde{\lambda}_t X_{A_t} + (1 - \tilde{\lambda}_t)X_{B_t}$$
$$\tilde{y} = \lambda y_A + (1 - \lambda)y_B, \tag{2}$$

FadeMixUp can be modeled for temporal variations and can learn temporally localizable feature without sharp boundary changes like other mixing algorithms. Since many videos include these overlapping effects at the scene change, FadeMixUp can be applied naturally.

Summarization of deleting, blending, and mixing data augmentation algorithms are described in Table 1. In the table, checkmark indicates the elements (pixels) can be changed along the spatial or temporal axis by augmentation methods. Compared to existing algorithms [10, 50, 52, 49], our proposed methods are extended temporally and spatio-temporally.

## 4    Experiments

### 4.1    Experimental Settings

For video action recognition, we train and evaluate on the UCF-101 [36] and HMDB-51 [25] dataset. UCF-101 originally consists of 13,320 videos with 101 classes. It consists of three train/test splits, but we used the modified split provided by the 1st VIPriors action recognition challenge that consists 4,795 training videos and 4,742 validation videos. HMDB-51 consists of 6,766 videos with 51 classes. We use original three train/test splits for training and evaluations.

Our experiments are trained and evaluated on a single GTX 1080-ti GPU and implemented by PyTorch framework. We use SlowFast-50 [11] as backbone network with 64 temporal frames because it is more lightweight and faster than other networks such as C3D [39], I3D [3], and S3D [47] without any pre-training and optical-flow. For baseline, basic data augmentation such as random crop with size 160, random scale jittering between [160, 200] for short side of video, and random horizontal flip is applied. For optimization, batch size is set to 16,

Table 2: Data Augmentation Results

| | Range | Top-1 Acc. | Top-5 Acc. |
|---|---|---|---|
| Baseline | | 49.37 | 73.62 |
| RandAugment | Spatial | 66.87 | 88.04 |
| | Temporal | 67.33 | 88.42 |
| | Temporal+ | **69.23** | **89.20** |
| | Mix | 68.24 | 89.25 |

Table 3: Data Deleting Results

| | Top-1 Acc. | Top-5 Acc. |
|---|---|---|
| Baseline | **49.37** | **73.62** |
| CutOut | 46.01 | 69.80 |
| FrameCutOut | 47.60 | 71.32 |
| CubeCutOut | 47.45 | 72.06 |

Table 4: Data Mixing Results

| | Top-1 Acc. | Top-5 Acc. |
|---|---|---|
| Baseline | 49.37 | 73.62 |
| CutMix($\alpha = 2$) | 50.81 | 75.62 |
| FrameCutMix($\alpha = 2$) | 51.29 | 74.99 |
| FrameCutMix($\alpha = 5$) | **53.10** | **76.61** |
| CubeCutMix($\alpha = 2$) | 51.86 | 74.34 |
| CubeCutMix($\alpha = 5$) | 51.81 | 75.16 |

Table 5: Data Blending Results

| | Top-1 Acc. | Top-5 Acc. |
|---|---|---|
| Baseline | 49.37 | 73.62 |
| MixUp | 59.60 | 82.56 |
| FadeMixUp | 59.22 | 82.24 |
| CutMixUp | 59.35 | 81.99 |
| FrameMixUp | **60.67** | **83.47** |
| CubeMixUp | 59.85 | 82.20 |

learning rate is set to 1e-4, weight decay of 1e-5 is used, learning rate warmup [14] and cosine learning rate scheduling [30] is used with Adam optimizer [22]. We train the all models for 150 epochs. For evaluation, we sample 10 clips uniformly along temporal axis, and average softmax predictions.

## 4.2 Data-level temporal data augmentations

Table 2 shows recognition results on UCF-101 validation set of VIPriors challenge. For all result tables, **bold** is the best one and underline is the second best . RandAugment-spatial indicates original implementation without temporal variations. In temporal version, M1 of Fig. 2 is sampled from $Uniform(0.1, M2)$ and M2 is set to M of spatial RandAugment. For temporal+, M1 and M2 are set to M$-\delta$ and M$+\delta$, respectively, where $\delta$ is sampled from $Uniform(0, 0.5 \times M)$. For Mix in Table 2, it randomly chooses spatial or temporal+. Results show that applying RandAugment solely improves recognition performance drastically. Among them, temporal expended RandAugment-T (temporal+) shows the best performance. For all RandAugment results, to produce the best accuracy, grid search of two hyper-parameters: N $\in [1, 2, 3]$ and M $\in [3, 5, 10]$, is used.

## 4.3 Data-level temporal deleting, mixing, and blending

For data deleting like CutOut [10], results of it and its temporal extensions, FrameCutOut and CubeCutOut, are described in Table 3. For CutOut, $80 \times 80$ spatial patch is randomly deleted, and for FrameCutOut, 16 frames are randomly deleted. For CubeCutOut $80 \times 80 \times 16$ cube is randomly deleted. Results show that deleting patches, frames, or spatio-temporal cubes hurts recognition performance in the limited number of dataset. Among them, CutOut shows the worst performances.

For data mixing like CutMix [50] and its extensions, results are described in Table 4. We apply the mixing probability of 0.5 for all methods and different

Table 6: Temporal Augmentation Results on HMDB51 Dataset

| | Split-1 | | Split-2 | | Split-3 | | Average | |
|---|---|---|---|---|---|---|---|---|
| | Top-1 Acc. | Top-5 Acc. | Top-1 Acc. | Top-5 Acc. | Top-1 Acc. | Top-5 Acc. | Top-1 Acc. | Top-5 Acc. |
| Baseline | 36.60 | 67.25 | 37.19 | 65.75 | 32.88 | 65.82 | 35.56 | 66.27 |
| RandAug | 47.45 | 79.21 | 47.12 | 76.86 | 47.45 | 77.97 | 47.34 | 78.01 |
| RandAug-T | **48.17** | **79.35** | **47.84** | **77.00** | **48.37** | **78.17** | **48.13** | **78.17** |
| CutOut | **34.71** | **65.49** | **32.35** | 63.79 | 31.76 | 62.94 | **32.94** | **64.07** |
| FrameCutOut | 31.05 | 61.57 | 32.16 | **65.36** | **31.87** | **64.18** | 31.69 | 63.70 |
| CubeCutOut | 33.01 | 63.99 | 32.04 | 64.25 | 30.59 | 62.81 | 31.88 | 63.68 |
| CutMix | 33.95 | 64.27 | 33.69 | 66.84 | 31.24 | 63.53 | 32.96 | 64.88 |
| FrameCutMix | 34.97 | **65.56** | 34.84 | **67.91** | 33.27 | 63.53 | 34.36 | 65.67 |
| CubeCutMix | **35.10** | 65.10 | **35.95** | 65.62 | **36.54** | **67.97** | **35.86** | **66.23** |
| MixUp | 38.95 | 68.10 | **40.72** | 70.92 | 40.20 | 71.31 | 39.96 | 70.11 |
| CutMixUp | **40.92** | **71.07** | 40.16 | 71.55 | 39.28 | 71.48 | 40.12 | 71.37 |
| FrameMixUp | 40.33 | 70.98 | 40.52 | 70.85 | 39.02 | 70.65 | 39.96 | 70.83 |
| CubeMixUp | 40.72 | 70.65 | 40.70 | **72.88** | **40.92** | **71.83** | **40.78** | **71.79** |
| FadeMixUp | 39.80 | 70.39 | 40.46 | 71.70 | 39.61 | 70.00 | 39.96 | 70.70 |

Table 7: Model Evaluation for VIPriors Challenge

| Train Data | Test Data | Augmentation | Regularization | Others | Top-1 Acc. | Top-5 Acc. |
|---|---|---|---|---|---|---|
| Train | Val | | | | 49.37 | 73.62 |
| Train | Val | RandAug-T | | | 69.23 | 89.20 |
| Train | Val | RandAug-T | FadeMixUp | | 68.73 | 89.27 |
| Train | Val | RandAug-T | FrameMixUp | | **69.70** | **89.84** |
| Train+Val | Test | | | | 68.99 | - |
| Train+Val | Test | RandAug-T | | | 81.43 | - |
| Train+Val | Test | RandAug-T | All Methods | Ensemble | **86.04** | - |

hyper-parameters $\alpha$. Since object size in the action recognition dataset is smaller than that of ImageNet [23], mixing ratio should be sampled in the region close to 0.5 by sampling large $\alpha$ in the beta distribution. Results show that the temporal and spatio-temporal extensions outperform spatial-only mixing strategy. Since probability of object occlusion is lower at temporal mixing than spatial mixing, FrameCutMix performance is the most improved.

Finally, for data blending, compared to MixUp [2] and CutMixUp [49], temporal and spatio-temporal extensions show slightly superior performance that are described in Table 5. Compared to deleting and mixing augmentations, blending shows the best performances. Since the number of training data is limited, linear convex combination of samples easily and effectively augments in the sample space.

## 4.4   Results on HMDB-51 dataset

To check the generalization to other dataset, we train and evaluate on HMDB-51 dataset with its original splits. Generally, recognition performance in HMDB-51

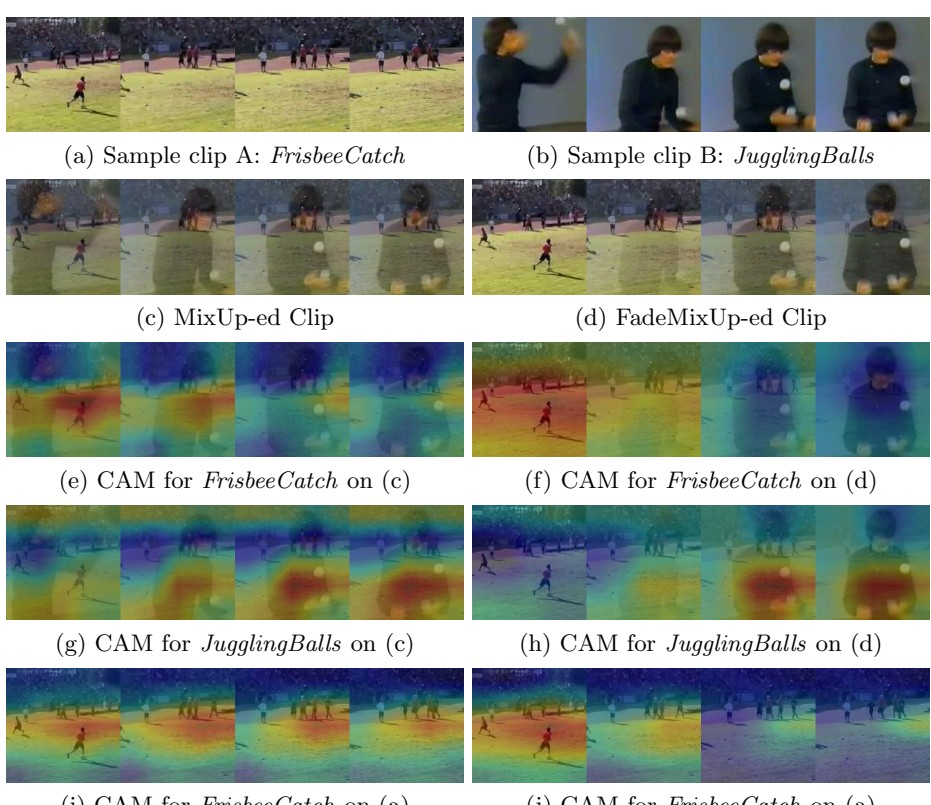

(a) Sample clip A: *FrisbeeCatch*          (b) Sample clip B: *JugglingBalls*

(c) MixUp-ed Clip          (d) FadeMixUp-ed Clip

(e) CAM for *FrisbeeCatch* on (c)          (f) CAM for *FrisbeeCatch* on (d)

(g) CAM for *JugglingBalls* on (c)          (h) CAM for *JugglingBalls* on (d)

(i) CAM for *FrisbeeCatch* on (a)          (j) CAM for *FrisbeeCatch* on (a)

Fig. 5: Class actionvation maps. *Left*: MixUp, *Right*: FadeMixUp

is inferior to the performance of UCF-101 due to its limited number of training samples. We use same model and hyper-parameters as in UCF-101.

Results in Table 6 show that temporal extensions generally outperforms spatial-only versions, and similar to UCF-101, RandAugment and blending methods show the best accuracies.

## 4.5   1st VIPriors action recognition challenge

Based on the comprehensive experimental results, we attend the 1st VIPriors action recognition challenge. In this challenge, any pre-training and using external dataset is not allowed. Performances on various models are described in Table 7. For validation, applying both RandAugment-T and FrameMixUp show the best result. For test set, total 3,783 videos are provided without ground truths. Therefore, we report the results based on the challenge leaderboard. Combination of training and validation dataset, total 9,537 videos are used for training the final challenge entries. From the baseline accuracy, 68.99%, adapting RandAugment-

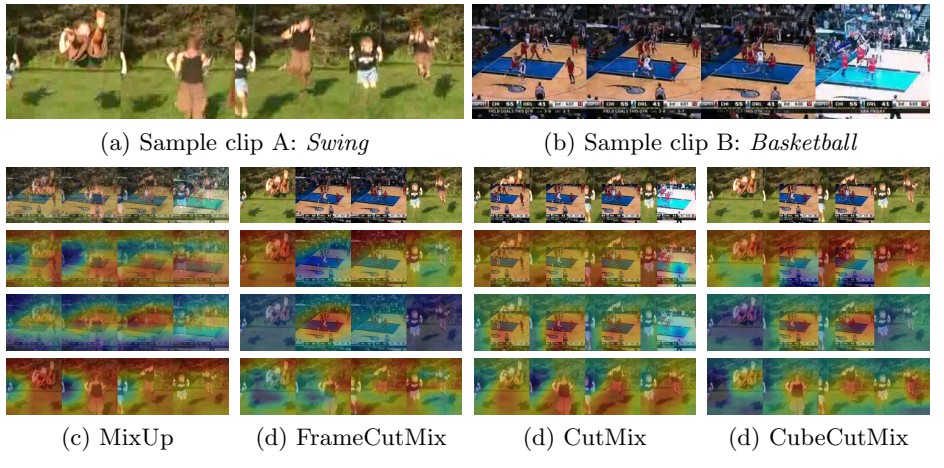

(a) Sample clip A: *Swing*                    (b) Sample clip B: *Basketball*

(c) MixUp        (d) FrameCutMix        (d) CutMix        (d) CubeCutMix

Fig. 6: Class actionvation maps. For (c)-(f), from the top to the bottom row: mixed clips, CAMs for *Swing*, CAMs for *Basketball*, and CAMs for *Swing* on pure clip (a), respectively.

T only improves the performance up to 81.43%. Finally, we submitted ensembled version of different models that are trained using RandAugment-T and various mixing and blending augmentations, to produce 86.04% Top-1 accuracy.

### 4.6   Discussions

**Why the improvement are not large?** Although the temporal extensions generally outperform spatial-only versions in data augmentation algorithms, the performance improvements might be not large enough. Possible reasons of this are three-fold, the first one is the lack of enough training data, and the second one is the lack of temporal perturbation, and the last one is the datasets are used for experiments consists trimmed videos. Both UCF-101 and HMDB-51 dataset have little temporal perturbations. Therefore, applying spatial augmentation is enough to learn the contexts. And both dataset are trimmed that have little temporal occlusions, which means there is no room to learn the ability to localize temporally. For deleting and mixing, compared to the image dataset, since the action region is relatively small, removing spatial region can hurts the basic recognition performance if the number of training data is not enough. In contrast, for blending, although it is unnatural image as said in [50], it can exploit full region of frames. Therefore it produces reasonable performance improvements.

**Spatio-temporal class activation map visualization** We visualize the learned feature using class activation map [55] in Fig. 5. In the SlowFast network, we use the feature of the last convolutional layer in SlowPath. Fig. 5 (a) and (b) are example clips. Fig. 5 (c) and (d) are the visualization of MixUp-ep and

FadeMixUp-ed clips, respectively. In Fig. 5 (f) and (h) compared to Fig. 5 (e) and (g), FadeMixUp features are more localized temporally than that of MixUp. In Fig. 5 (j) compared to Fig. 5 (i), activations of FadeMixUp is spatio-temporally localized better than that of MixUp in the pure clip A.

Fig. 6 compares spatio-temporal localization abilities between MixUp, Cut-Mix, FrameCutMix, and CubeCutMix. Compared to MixUp, as said in their paper [50], CutMix can localize spatially for basketball field and the person on swing. However, compared to CubeCutMix, activations of CutMix is not localized temporally well. FrameCutMix also cannot localize feature like MixUp, but it can separate the weights of activation separately in temporal axis.

## 5    Conclusion

In this paper, we proposed several extensions of data-level augmentation and data-level deleting, blending, and mixing augmentation algorithms from the spatial, or image domain into temporal and spatio-temporal, or video domain. Although applying spatial data augmentation itself increases the recognition performance in a limited amount of dataset, extending temporal and spatio-temporal data augmentation boosts the performance. Moreover, our models trained on temporal augmentation have abilities to localize temporally and spatio-temporally that cannot be achieved from the model trained on spatial augmentations only. Our next step will be an extension to the large-scale dataset such as Kinetics [3], or untrimmed videos.

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
