# OpenReview forum: "Learning Temporally Invariant and Localizable Features via Data Augmentation for Video Recognition"
_thecvf.com/ECCV/2020/Workshop/VIPriors — VIPriors Oral_

### Official Review · AnonReviewer1 · 2020-07-21
**Good execution of a simple idea**

**Confidence:** 3
**Rating:** 9

**Review:**

[Summary] In 2-3 sentences, describe the key ideas, experiments, and their significance.

The authors extend popular data augmentation methods to the temporal domain in a straightforward manner. Experiments show minor improvements over the spatial data augmentations.

[Strengths] What are the strengths of the paper? Clearly explain why these aspects of the paper are valuable.

Simple, powerful idea; simple implementations; clear explanations; self-critical in analyzing the magnitude of their contribution; extensive evaluations.

[Weaknesses] What are the weaknesses of the paper? Clearly explain why these aspects of the paper are weak.

Marginally related to workshop topic.

[Overall rating] Paper rating: Strong accept

[Detailed comments] Additional comments regarding the paper (e.g. typos or other possible improvements you would like to see for the camera-ready version of the paper, if any.)

- Grammar: lines 40, 254, 521 (whole paragraph needs revision)
- Typos: line 130 "squre", 141 "researches"
- Figure 2 could have been pseudocode

---

### Official Review · AnonReviewer2 · 2020-07-28
**Simple, effective and very well discussed**

**Confidence:** 4
**Rating:** 7

**Review:**

1. [Summary] In 2-3 sentences, describe the key ideas, experiments, and their significance.

 This paper proposes to adapt several data-level augmentation techniques from image field to videos. To study the effects of such techniques, authors conduct experiments on the action recognition topic. Results showcase that some of the proposed techniques helps improve the performance. Additionally, they described their participation in the 1st VIPriors Action Recognition Challenge.

2. [Strengths] What are the strengths of the paper? Clearly explain why these aspects of the paper are valuable.

 -	Regarding the results, the paper is really well motivated and clearly understandable. The story is clear and is very easy to follow.
 -	Authors clearly described how they have adapted all the techniques to video.
 -	The results are marginal (or even worse than baseline) in some situations. However, I like the fact that authors recognise it and discuss it in Section 4.6, suggesting some interesting reasons.

3. [Weaknesses] What are the weaknesses of the paper? Clearly explain why these aspects of the paper are weak.

 -	Results are marginal (or even worse than baseline) for some augmentation techniques.
 -	Authors talk always about temporal distortions. However, it seems that basically they apply frame distortions during a concrete temporal window. For me, this is not a temporal distortion. Authors can check the paper “Learning Temporal Action Proposals With Fewer Labels”. In this paper, data augmentation on videos is performed by modifying the temporal information accumulated by the features. Concretely, they use time warping and Time masking.

4. [Overall rating] Paper rating.

 7

5. [Justification of rating] Please explain how the strengths and weaknesses aforementioned were weighed in for the rating.

 Despite weaknesses, the paper is well written, and very well discussed.

6. [Detailed comments] Additional comments regarding the paper (e.g. typos or other possible improvements you would like to see for the camera-ready version of the paper, if any.)

 - Line 198: augmentation operation(s).
 - Line 350: consists (of)

---

### Decision · Program_Chairs · 2020-07-29

**Decision:**

Accept (Oral)

**Comment:**

It is our pleasure to inform you that your paper has been accepted to the oral track of the 1st Visual Inductive Priors for Data-Efficient Deep Learning Workshop.

Please note the following deadlines:
* August 11, 2020 - workshop material, including:
 * paper in PDF format;
 * pre-recorded video presentation;
 * slides of the presentation in PDF.
* September 15, 2020 - camera-ready paper

The reviews can be found on OpenReview. Please take these comments and suggestions into account when preparing the camera-ready version of your paper, which is due September 15, 2020. The camera-ready paper should be uploaded to OpenReview.

As part of the workshop, each paper for oral presentation must submit a pre-recorded 5 minute talk before August 11, 2020. You will receive more information on how to upload the material shortly. The requirements for the video are:
* Duration: maximum 5 minutes
* MP4 format
* File size max. 100 MB
* Has an inset with a video of the speaker
* 16:9 aspect ratio (strongly preferred)
* 1920x1080 resolution (strongly preferred, at least 720 height)

Our suggested software for pre-recording your presentation is Zoom. For more information, please refer to the following guides:
How to record with Zoom Guide: http://homepages.inf.ed.ac.uk/rbf/ECCV2020HowtoRecordusingZoom.pdf
How to Record with Zoom tutorial: https://www.youtube.com/watch?v=CR199W7HdC0
Please ensure that at least one of the authors of the paper is available to attend the workshop during the allotted times. Note that the workshop will take place in two sessions spread across time zones (details are to follow). We will send instructions on how to connect to the workshop as soon as possible. The schedule for all talks and papers will be posted soon at the workshop website: https://vipriors.github.io.

We look forward to seeing you at the workshop!